# Silage Quality and Output of Different Maize–Soybean Strip Intercropping Patterns

Tairu Zeng [1,†], Yushan Wu [2,†], Yafen Xin [1], Chen Chen [1], Zhaochang Du [1], Xiaoling Li [1], Junfeng Zhong [1], Muhammad Tahir [1], Bo Kang [3], Dongmei Jiang [3], Xiaochun Wang [2], Wenyu Yang [2] and Yanhong Yan [1,*]

[1] College of Grassland Science and Technology, Sichuan Agricultural University, Chengdu 611130, China; 2018302004@stu.sicau.edu.cn (T.Z.); 2020302093@stu.sicau.edu.cn (Y.X.); 2020202076@stu.sicau.edu.cn (C.C.); 2019202038@stu.sicau.edu.cn (Z.D.); s20175506@stu.sicau.edu.cn (X.L.); 201804561@stu.sicau.edu.cn (J.Z.); memmmerani@stu.sicau.edu.cn (M.T.)

[2] College of Agronomy, Sichuan Agricultural University, Chengdu 611130, China; yushan.wu@sicau.edu.cn (Y.W.); xchwang@sicau.edu.cn (X.W.); mssiyangwy@sicau.edu.cn (W.Y.)

[3] College of Animal Science and Technology, Sichuan Agricultural University, Chengdu 611130, China; bokang@sicau.edu.cn (B.K.); jiangdm@sicau.edu.cn (D.J.)

[*] Correspondence: yanyanhong3588284@126.com

[†] These authors contributed equally to this work.

**Abstract:** Intercropping improves land-use efficiency under conditions of limited land and resources, but no information is currently available pertaining to land-use efficiency and silage quality based on whole-plant utilization. Therefore, a two-year field experiment was conducted with the following conditions: three maize–soybean strip intercropping patterns (SIPs), comprising two maize rows along with two, three, or four soybean rows (2M2S, 2M3S, and 2M4S, respectively); and two sole cropping patterns of maize (SM) and soybean (SS). The aim was to evaluate the biomass yield and silage quality under each condition. Our results showed that all SIPs had a land equivalent ratio (*LER*) of over 1.6 based on both fresh and dry matter yield, and a higher whole plant yield, compared to sole cropping. Specifically, 2M3S exhibited the highest whole crop dry matter *LER* (1.8–1.9) and yield (24.6–27.2 t ha$^{-1}$) compared to SM and SS (20.88–21.49 and 3.48–4.79 t ha$^{-1}$, respectively). Maize–soybean mixed silages also showed better fermentation quality with higher lactic acid content (1–3%) and lower ammonia-N content (2–8%) compared to SS silages, and higher crude protein content (1–1.5%) with lower ammonia-N content (1–2%) compared to SM silage. Among the intercropping patterns, 2M3S had the highest fermentation quality index V-score (92–95). Consequently, maize–soybean strip intercropping improved silage quality and biomass yield, with 2M3S being recommended, due to its highest *LER* and biomass yield, and most optimal silage quality.

**Keywords:** row ratio of maize and soybean; land equivalent ratio; biomass yield; silage

## 1. Introduction

As the demand for meat is ever-increasing with the rapid development of the world [1], silage is becoming increasingly important. It offers livestock feed during the non-growth stage of forage, thereby guaranteeing a year-round feed supply [2]. Maize (*Zea mays* L.) silage is prevalent due to its low buffering capacity and high content of water-soluble carbohydrates, making it easy to ensile into energy-rich feed [3], but its protein content is insufficient. Soybean (*Glycine max Linn*. Merr.) silage, on the other hand, is rich in protein and vitamins but is prone to spoil, and produce unpleasant odors due to the effects of butyric acid [4,5]. A combination of maize and soybean silage has proven successful, as the plentiful carbohydrates supplied by the maize provide an adequate substrate for lactic acid bacteria to proliferate, thereby guaranteeing quality fermentation, whilst the high protein content of the soybean improves the nutritional content [6,7]. Parra et al. [8] obtained higher protein contents with maize and soybean mixed silage at different soybean weight

proportions versus maize silage alone, and also found the mixed silage to be well-fermented (indicated by a pH lower than 4.0). However, in this previous study, conventional corn and soybean were planted and harvested separately and then ensiled at a certain weight ratio. If maize and soybean were planted and harvested simultaneously then they could be ensiled directly according to the planting patterns instead of the weight ratio, potentially saving labor, cost, and space.

Intercropping is a classic and sustainable agricultural practice that grows multiple crops in the same field [9], and it improves biomass yield and resource utilization rates [10]. Maize–soybean strip intercropping has been proven to increase resource utilization rates and crop yield [11], and the land equivalent ratio [12]. Liu et al. [13] did report achieving a land equivalent ratio of 1.4 in maize–soybean intercropping based on grain yield. Whereas, previous studies have barely reported the land equivalent ratio for whole crop biomass yield. Batista et al. [14] reported that the same numbers of maize and soybean rows showed a higher crude protein yield and similar maize grain yield under intercropping versus sole maize cropping conditions. To sum up, intercropping greatly affects crop yield and quality [15], but how intercropping affects silage quality and whole crop biomass yield remains unknown.

Therefore, the objective of this study was to investigate: (1) whole crop biomass yield in different strip intercropping patterns; (2) silage quality in different strip intercropping patterns; (3) the correlation between the biomass yield and the silage quality.

## 2. Materials and Methods

### 2.1. Experimental Site

A two-year field experiment was conducted during 2018 and 2019 at Chongzhou experimental farm of Sichuan Agricultural University (30°33′ N, 103°38′ E, altitude 556 m). The climate conditions of the current study were subtropical with high humidity. The experimental site had an annual mean rainfall of 969 mm and a temperature of 16.08 °C.

The soil had a pH of 6.30, organic matter of 37.6 g kg$^{-1}$, total N of 2.03 g kg$^{-1}$, available N of 135.7 mg kg$^{-1}$, available P of 10.2 mg kg$^{-1}$, and available K of 101.1 mg kg$^{-1}$ in the topsoil layer (0–20 cm).

### 2.2. Experimental Design

The field experiment was a randomized complete block design with triplicates. This study utilized one silage maize cultivar (Yayu 04889, 98 days from seedling emergence to silage harvest period, medium-early maturing variety), one crop soybean cultivar (Nandou 25, 102 days from seedling emergence to silage harvest period, medium-early maturing variety, $S_1$), and one forage soybean cultivar (Fendoumulv 2, 106 days from seedling emergence to silage harvest period, medium maturing variety, $S_2$). There were five different planting patterns (Figure 1), i.e., sole cropping maize (SM), sole cropping soybean (SS), two maize rows plus two soybean rows (2M2S), two maize rows plus three soybean rows (2M3S), and two maize rows plus four soybean rows (2M4S). The cultivation arrangement is shown in Table S1. Seeding of sole cropping crops was conducted in line with the local planting densities: 75,000 plants ha$^{-1}$ for maize and 150,000 plants ha$^{-1}$ for soybean. The same planting density was used in 2M2S, 2M3S, and 2M4S. All agronomic procedures, i.e., sowing, harvesting, and weeding, were done manually.

Maize and soybean were sown synchronously on 17 April 2018 and 24 April 2019, and harvested simultaneously on 29 July 2018, and 10 August 2019, respectively. The corresponding stage of maize was the 2/3 milk line stage, and the stages for soybean $S_1$ and $S_2$ were R6 (seed filling period) and R5 (early seed filling period), respectively.

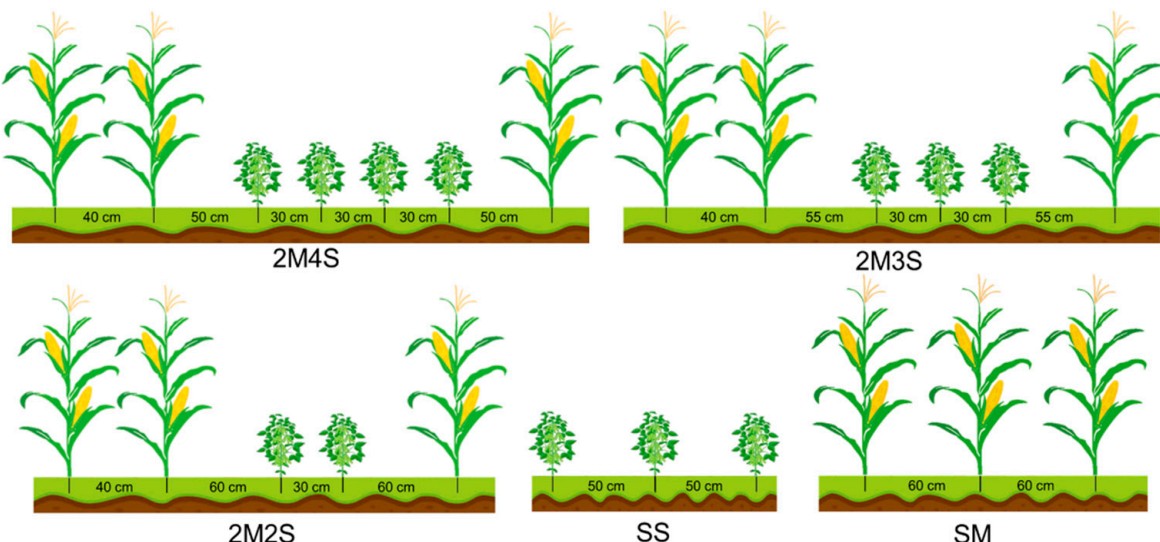

**Figure 1.** Different strip intercropping patterns of maize–soybean. Row spacings for soybean and maize rows in all strip intercropping planting patterns were 0.30 m and 0.40 m, respectively. The other spatial parameters were as follows: for 2M2S, the strip length was 1.90 m with 0.60 m distance between rows of maize and soybean, and the column spacing was 14.03 cm for maize and 7.01 cm for soybean; for 2M3S, the strip length was 2.10 m with 0.55 m distance between rows of maize and soybean, and the column spacing was 12.70 cm for maize and 9.52 cm for soybean; for 2M4S, the distance between the rows of soybean and maize was 0.50 m with a total strip length of 2.30 m, and the column spacing was 11.59 cm for maize and 11.59 cm for soybean; for SM, the distance between rows of sole cropping maize was 0.60 m, and the column spacing was 0.23 m; for SS, the distance between soybean rows was of 0.50 m, and the column spacing was 0.13 m.

### 2.3. Sampling and Measurement

### 2.3.1. Total Crop Yield

According to the ratio of maize and soybean planting density, 6 maize and 12 soybean whole crop plants were harvested manually with a stubble height of 15 cm above ground. The plants harvested for each condition were as follows: 2M2S (3 maize plants for each row and 6 soybean plants for each row), 2M3S (3 maize plants for each row and 4 soybean plants for each row), 2M4S (3 maize plants for each row and 3 soybean plants for each row), SS (4 plants for each row), and SM (3 plants for each row). Then, samples were weighed immediately for whole crop fresh matter yield (WFM, t ha$^{-1}$) and 1 kg of matter was subjected to air-forced oven drying for 1 h at 105 °C, and then at 65 °C until the weight became constant, to determine whole crop dry matter (DM). The DM content was then used to calculate whole crop dry matter yield as planting density (WDM, t ha$^{-1}$).

### 2.3.2. Competition Parameters

The land equivalent ratio (*LER*) was analyzed to determine the yield advantage given by different maize–soybean SIPs [16], as follows:

$$LER = \left(\frac{Ymb}{Ym}\right) \times \left(\frac{Ysb}{Ys}\right) = LERm + LERs \qquad (1)$$

where *Ymb* and *Ym* are the yield of maize under SM and SIP, respectively; *Ysb* and *Ys* are the yield of soybean under SS and SIP, respectively; and *LERs* and *LERm* are the *LER* of soybean and maize, respectively.

The competition ratio (*CR*) was utilized to assess the competition between maize and soybean in each SIP. The calculation of *CR* according to Mead and Willey [16] was as follows:

$$CRm = \left(\frac{LERm}{LERs}\right) \times \left(\frac{Zs}{Zm}\right) \qquad (2)$$

$$CRm = \left(\frac{LERm}{LERs}\right) \times \left(\frac{Zs}{Zm}\right) \qquad (3)$$

where *Zm* and *Zs* are the sown proportion of maize and soybean in the SIP, respectively.

### 2.4. Silage Preparation

The silage experiment was only conducted in 2018. The maize and soybean were sampled according to the ratio of maize and soybean planting density, meaning the specific sampling was the same as that of the total crop yield (2.3.1). Then, the samples were manually chopped into 20 mm sections and immediately packed into polyethylene plastic bags (25 × 35 cm, Aodeju, Sichuan, China), which were then sealed with a vacuum sealer (evox-30, Orved Spa., Musile di Piave, Italy) in triplicate, before being stored at room temperature (25–30 °C) for 60 days (d). Each bag was loaded with 300 g of the sample. A total of 27 disposable sample bags were filled [(3 intercropping conditions + SS condition) × 2 soybean varieties + SM condition] × 3 replications. After 60 days, the sample bags were opened to analyze fermentation and nutritional quality.

### 2.5. Silage Profiles Determination

After 60 days of ensiling, 20 g of the silage sample was taken and placed in 180 mL of sterilized water. The mixture was homogenized by mixing at 4 °C for 24 h, then the water was extracted to enable pH and organic acid determination, according to Yan et al. [17]. Lactic acid (LA), acetic acid (AA), butyric acid (BA), and propionic acid (PA) were determined using HPLC, according to Zeng et al. [5]. Dry matter content determination of pre- and post-ensiling samples followed the same procedure described in Section 2.3.1. Dried samples were then ground through a 1 mm screen for chemical component analysis. The water-soluble carbohydrate (WSC) content was determined using the thracenone-sulphuric acid method [18]. Ammonia nitrogen ($NH_3$-N) was determined according to Broderick and Kang [19]. Crude protein (CP) content was measured using the Kjeldahl method [18]. The neutral detergent fiber (NDF) content and acid detergent fiber (ADF) contents were assayed according to Van Soest et al. [20]. Relative feed value (RFV) and V-score, the indices to estimate the chemical and fermentation quality of feedstock, were calculated using the methods of Van Dyke and Anderson [21] and Cao et al. [22], respectively.

### 2.6. Statistical Analysis

All the data were processed following a two-way analysis of variance using SPSS 23.0 software (SPSS Inc., Chicago, IL, USA). Duncan's multiple comparisons were used for different sample means and significance was determined at the 5% and 1% levels. The heatmap was processed using OriginLab 2022 (https://www.originlab.com/, accessed on 28 February 2022) with a Pearson correlation coefficient ranging from −1 to 1.

## 3. Results

### 3.1. Field Profiles

As is shown in Tables 1 and 2, all yield indices were affected by maize–soybean planting patterns with the same trend in both years ($p < 0.01$). The maize fresh matter (MFM) yield generally decreased with the increase in soybean rows, with 2M2S exhibiting the highest MFM yield (63.50–65.23 t ha$^{-1}$), and 2M4S exhibiting a MFM yield lower than that of sole maize (SM) cropping. In addition, the maize dry matter (MDM) yield was the highest in 2M3S (21.71–21.90 t ha$^{-1}$). The soybean fresh matter (SFM) yield (13.20–14.06 t ha$^{-1}$) and dry matter (SDM) (4.10–4.26 t ha$^{-1}$) yield were affected by planting patterns and soybean varieties ($p < 0.01$), with the soybean sole (SS) cropping showing the highest FM and DM

yield, and no differenced found among the intercropping treatments. The 2M2S and 2M3S conditions displayed higher maize–soybean fresh matter (MSFM) yield (70.30–75.60 t ha$^{-1}$), while 2M3S had the highest maize–soybean dry matter (MSDM) yield (24.96–27.20 t ha$^{-1}$).

The *LER* of the FM yield and the DM are displayed in Table 3. All the *LER* indices were affected by planting patterns in both years ($p \leq 0.01$). The fresh matter *LERm* (*FLERm*) was the highest in 2M2S (1.05–1.06) and generally decreased with the increase in soybean rows, while the fresh matter *LERs* (*FLERs*) displayed an opposite trend, with 2M4S exhibiting the highest *FLERs* level (0.82–0.84). The highest total fresh matter *LER* (FLER) was higher for 2M3S and 2M4S than for 2M2S. The highest dry matter *LERm* (*DLERm*) was found in 2M3S (1.04–1.10), while that of the 2M4S was the lowest (0.92–1.02). Moreover, the dry matter *LERs* (*DLERs*) and the total dry matter *LER* (*DLER*) showed a similar trend with regards to *FLERs* and *FLER*, with the 2M4S and 2M3S exhibiting the highest levels (0.81–0.93 and 1.84–1.93, respectively).

As shown in Figure 2, in both years, the fresh and dry matter *CRm* decreased with the increase in soybean rows, with the highest level observed for 2M2S (0.70–0.76). Meanwhile, the *CRs* displayed an opposite trend, with 2M4S exhibiting the highest level (0.41–0.46) ($p < 0.05$). The fresh and dry matter *CRm* were always higher than those of the *CRs*.

**Table 1.** Whole crop fresh and dry matter yield of maize and soybean as affected by different planting patterns during 2018.

| Treatment 2018 | MFM t ha$^{-1}$ | SFM t ha$^{-1}$ | MSFM t ha$^{-1}$ | MDM t ha$^{-1}$ | SDM t ha$^{-1}$ | MSDM t ha$^{-1}$ |
|---|---|---|---|---|---|---|
| 2M2S$_1$ | 65.26 | 10.34 | 75.60 | 23.36 | 3.40 | 26.76 |
| 2M3S$_1$ | 63.05 | 11.84 | 74.90 | 23.23 | 3.98 | 27.20 |
| 2M4S$_1$ | 59.39 | 12.58 | 71.98 | 21.58 | 4.17 | 25.75 |
| 2M2S$_2$ | 65.20 | 7.65 | 72.85 | 23.43 | 2.45 | 25.87 |
| 2M3S$_2$ | 62.51 | 8.89 | 71.4 | 23.58 | 2.90 | 26.48 |
| 2M4S$_2$ | 59.26 | 9.60 | 68.86 | 21.85 | 3.25 | 25.09 |
| SS$_1$ | - | 14.94 | 14.94 | - | 4.73 | 4.73 |
| SS$_2$ | - | 11.46 | 11.46 | - | 3.48 | 3.48 |
| SM | 62.49 | - | 62.49 | 21.49 | - | 21.49 |
| SEM | 1.41 | 0.35 | 3.29 | 1.25 | 0.18 | 1.44 |
| V means | | | | | | |
| S$_1$ | 62.57 | 12.43 [a] | 59.35 [a] | 22.72 | 4.07 [a] | 21.11 |
| S$_2$ | 62.33 | 9.40 [b] | 56.14 [b] | 22.95 | 3.02 [b] | 20.23 |
| P means | | | | | | |
| 2M2S | 65.23 [a] | 8.99 [b] | 74.22 [a] | 23.39 [a] | 2.93 [c] | 26.32 [a] |
| 2M3S | 62.78 [b] | 10.37 [b] | 73.15 [a] | 23.40 [a] | 3.44 [bc] | 26.84 [a] |
| 2M4S | 59.33 [c] | 11.09 [b] | 70.42 [b] | 21.71 [b] | 3.71 [b] | 25.42 [b] |
| SM | 62.49 [b] | 0 | 62.49 [c] | 21.49 [b] | 0 | 21.49 [c] |
| SS | 0 | 13.20 [a] | 13.20 [d] | 0 | 4.10 [a] | 4.10 [d] |
| Significance | | | | | | |
| V | 0.78 | <0.01 | <0.01 | 0.44 | <0.01 | 0.08 |
| P | <0.01 | <0.01 | <0.01 | <0.01 | <0.01 | <0.01 |
| V$^*$P | 0.13 | 0.6 | 0.21 | 0.93 | 0.33 | 0.31 |

S$_1$, treatments containing crop soybean; S$_2$, treatments containing forage soybean; 2M2S, 2 maize rows plus 2 soybean rows strip intercropping; 2M3S, 2 maize rows plus 3 soybean rows strip intercropping; 2M4S, 2 maize rows plus 4 soybean rows strip intercropping; SS$_1$, crop soybean sole cropping; SS$_2$, forage soybean sole cropping; SM, maize sole cropping. MFM, maize fresh matter yield; SFM, soybean fresh matter yield; MSFM, maize and soybean fresh matter yield; MDM, maize dry matter yield; SDM, soybean dry matter yield; MSDM, maize and soybean dry matter yield; V, main effect of soybean varieties; P, main effect of planting patterns; V*P, interactive effect between soybean varieties and planting patterns. SEM, standard error of means. Means are averaged over three replicates. Means that do not share the same letters in the column differ significantly ($p < 0.05$).

**Table 2.** Whole crop fresh and dry matter yield of maize and soybean as affected by different planting patterns in 2019.

| Treatment 2019 | MFM t ha$^{-1}$ | SFM t ha$^{-1}$ | MSFM t ha$^{-1}$ | MDM t ha$^{-1}$ | SDM t ha$^{-1}$ | MSDM t ha$^{-1}$ |
|---|---|---|---|---|---|---|
| 2M2S$_1$ | 63.40 | 11.14 | 74.54 [a] | 20.75 | 3.42 | 24.17 |
| 2M3S$_1$ | 60.84 | 12.44 | 73.29 [a] | 21.77 | 3.90 | 25.68 |
| 2M4S$_1$ | 57.32 | 13.38 | 70.70 [b] | 19.66 | 4.06 | 23.72 |
| 2M2S$_2$ | 63.61 | 7.15 | 70.76 [b] | 21.24 | 2.25 | 23.49 |
| 2M3S$_2$ | 61.37 | 8.93 | 70.30 [b] | 22.02 | 2.94 | 24.96 |
| 2M4S$_2$ | 57.38 | 9.70 | 67.08 [c] | 19.29 | 3.00 | 22.3 |
| SS$_1$ | - | 15.83 | 15.83 [f] | - | 4.79 | 4.79 |
| SS$_2$ | - | 12.29 | 12.29 [j] | - | 3.73 | 3.73 |
| SM | 60.10 | - | 60.10 [e] | 20.88 | - | 20.88 |
| SEM | 2.96 | 0.85 | 2.11 | 1.03 | 0.02 | 1.6 |
| V means | | | | | | |
| S$_1$ | 60.52 | 13.20 [a] | 58.59 | 20.73 | 4.04 [a] | 19.59 [a] |
| S$_2$ | 60.79 | 9.51 [b] | 55.11 | 20.85 | 2.98 [b] | 18.62 [b] |
| P means | | | | | | |
| 2M2S | 63.50 [a] | 9.15 [b] | 72.65 | 20.99 [b] | 2.83 [b] | 23.83 [b] |
| 2M3S | 61.11 [b] | 10.69 [b] | 71.79 | 21.90 [a] | 3.42 [b] | 25.32 [a] |
| 2M4S | 57.35 [d] | 11.54 [b] | 68.89 | 19.48c | 3.53 [b] | 23.01 [c] |
| SS | 0 | 14.06 [a] | 14.06 | 0 | 4.26 [a] | 4.26 [e] |
| SM | 60.10 [c] | 0 | 60.10 | 20.88 [b] | 0 | 20.88 [d] |
| Significance | | | | | | |
| V | 0.51 | <0.01 | <0.01 | 0.58 | <0.01 | <0.01 |
| P | <0.01 | <0.01 | <0.01 | <0.01 | <0.01 | <0.01 |
| V*P | 0.92 | 0.70 | <0.01 | 0.31 | 0.56 | 0.28 |

S$_1$, treatments containing crop soybean; S$_2$, treatments containing forage soybean; 2M2S, 2 maize rows plus 2 soybean rows strip intercropping; 2M3S, 2 maize rows plus 3 soybean rows strip intercropping; 2M4S, 2 maize rows plus 4 soybean rows strip intercropping; SS$_1$, crop soybean sole cropping; SS$_2$, forage soybean sole cropping; SM, maize sole cropping. MFM, maize fresh matter yield; SFM, soybean fresh matter yield; MSFM, maize and soybean fresh matter yield; MDM, maize dry matter yield; SDM, soybean dry matter yield; MSDM, maize and soybean dry matter yield; V, main effect of soybean varieties; P, main effect of planting patterns; V*P, interactive effect between soybean varieties and planting patterns. SEM, standard error of means. Means are averaged over three replicates. Means that do not share the same letters in the column differ significantly ($p < 0.05$).

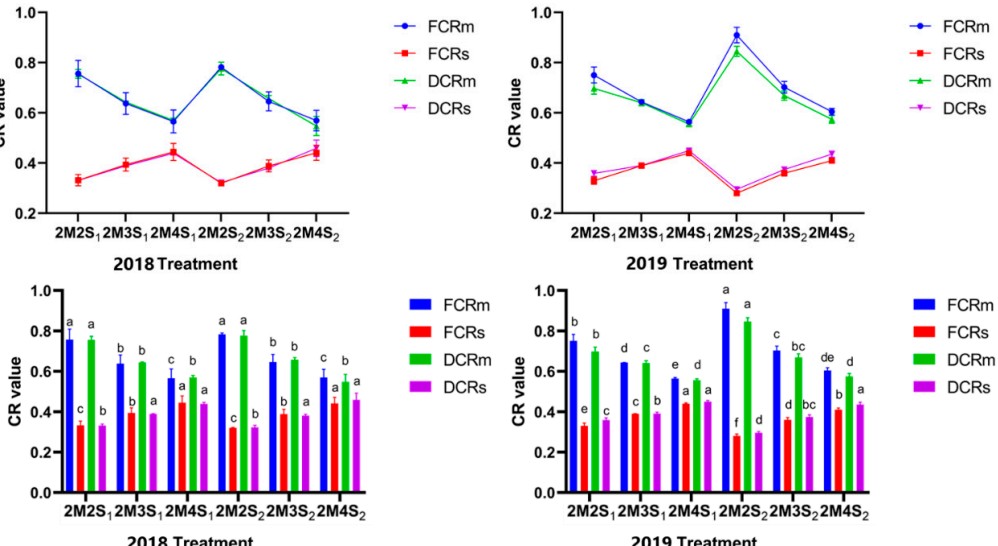

**Figure 2.** Competition ratios of maize–soybean intercropping. *FCRm*, competition ratio of maize based on fresh matter yield; *FCRs*, competition ratio of soybean based on fresh matter yield; *DCRm*, competition ratio of maize based on dry matter yield; *DCRs*, competition ratio of soybean based on dry matter yield. Different lowercase letters above the column indicate significant differences ($p < 0.05$).

**Table 3.** Land equivalent ratios of maize and soybean as affected by different planting patterns in 2019.

| Treatment 2018 | FLERm | FLERs | Total FLER | DLERm | DLERs | Total DLER | Treatment 2019 | FLERm | FLERs | Total FLER | DLERm | DLERs | Total DLER |
|---|---|---|---|---|---|---|---|---|---|---|---|---|---|
| 2M2S$_1$ | 1.05 | 0.69 | 1.74 | 1.09 | 0.72 | 1.80 | 2M2S$_1$ | 1.05 | 0.70 | 1.76 | 0.99 [c] | 0.71 [c] | 1.71 |
| 2M3S$_1$ | 1.01 | 0.79 | 1.80 | 1.08 | 0.84 | 1.92 | 2M3S$_1$ | 1.01 | 0.79 | 1.80 | 1.04 [a] | 0.81 [ab] | 1.86 |
| 2M4S$_1$ | 0.95 | 0.84 | 1.80 | 1.01 | 0.88 | 1.89 | 2M4S$_1$ | 0.95 | 0.85 | 1.80 | 0.94 [d] | 0.85 [a] | 1.79 |
| 2M2S$_2$ | 1.04 | 0.67 | 1.71 | 1.09 | 0.70 | 1.79 | 2M2S$_2$ | 1.06 | 0.58 | 1.64 | 1.02 [b] | 0.60 [d] | 1.62 |
| 2M3S$_2$ | 1.00 | 0.78 | 1.78 | 1.10 | 0.84 | 1.93 | 2M3S$_2$ | 1.02 | 0.73 | 1.75 | 1.05 [a] | 0.79 [b] | 1.84 |
| 2M4S$_2$ | 0.95 | 0.84 | 1.79 | 1.02 | 0.93 | 1.95 | 2M4S$_2$ | 0.95 | 0.79 | 1.74 | 0.92 [d] | 0.81 [ab] | 1.73 |
| SEM | 0.01 | 0.01 | 0.02 | 0.02 | 0.01 | 0.02 | SEM | 0.01 | 0.02 | 0.02 | 0.01 | 0.02 | 0.02 |
| V means | | | | | | | V means | | | | | | |
| S$_1$ | 1.00 | 0.78 | 1.78 | 1.06 | 0.81 | 1.87 | S$_1$ | 1.01 | 0.78 [a] | 1.79 [a] | 0.99 | 0.79 | 1.78 [a] |
| S$_2$ | 1.00 | 0.76 | 1.76 | 1.07 | 0.82 | 1.89 | S$_2$ | 1.01 | 0.70 [b] | 1.71 [b] | 1.00 | 0.73 | 1.73 [b] |
| P means | | | | | | | P means | | | | | | |
| 2M2S | 1.05 [a] | 0.68 [b] | 1.73 [b] | 1.09 [a] | 0.71 [c] | 1.80 [b] | 2M2S | 1.06 [a] | 0.64 [b] | 1.70 [b] | 1.01 | 0.66 | 1.66 [c] |
| 2M3S | 1.01 [a] | 0.79 [a] | 1.79 [a] | 1.09 [a] | 0.84 [b] | 1.93 [a] | 2M3S | 1.02 [b] | 0.76 [a] | 1.77 [a] | 1.05 | 0.80 | 1.85 [a] |
| 2M4S | 0.95 [b] | 0.84 [a] | 1.80 [a] | 1.01 [b] | 0.91 [a] | 1.92 [a] | 2M4S | 0.95 [c] | 0.82 [a] | 1.77 [a] | 0.93 | 0.83 | 1.76 [b] |
| Significance | | | | | | | Significance | | | | | | |
| V | 0.86 | 0.56 | 0.33 | 0.53 | 0.48 | 0.39 | V | 0.28 | <0.01 | <0.01 | 0.26 | <0.01 | <0.01 |
| P | 0.01 | <0.01 | <0.01 | <0.01 | <0.01 | <0.01 | P | <0.01 | <0.01 | <0.01 | <0.01 | <0.01 | <0.01 |
| V*P | 0.99 | 0.95 | 0.86 | 0.95 | 0.11 | 0.36 | V*P | 0.72 | 0.07 | 0.12 | 0.02 | 0.02 | 0.13 |

S1, treatments containing crop soybean; S2, treatments containing forage soybean; *FLERm*, land equivalent ratio of maize fresh matter; *FLERs*, land equivalent ratio of soybean fresh matter; Total *FLER*, land equivalent ratio of maize and soybean fresh matter; *DLERm*, land equivalent ratio of maize dry matter; *DLERs*, land equivalent ratio of soybean dry matter; Total *DLER*, land equivalent ratio of maize and soybean dry matter; V, main effect of soybean varieties; P, main effect of planting patterns; V*P, interactive effect between soybean varieties and planting patterns. SEM, standard error of means. Means are averaged over three replicates. Means that do not share the same letters in the column differ significantly ($p < 0.05$).

### 3.2. Chemical Composition of the Maize and Soybean

As shown in Table 4, all indices of the chemical composition of maize and soybean were affected by the interaction of planting patterns and soybean varieties ($p < 0.01$), except for the crude protein and ADF content. The WDM content of 2M3S2 and 2M4S2 was correspondingly higher than that of S$_1$, and the WDM content of the intercropped plants was higher than those in the sole cropping conditions. The highest WDM was observed for 2M3S (36–37%). The 2M2S$_2$ condition produced the lowest WDM content for S$_2$, while there was no significant difference between 2M2S$_1$ and 2M4S$_1$. The WSC content of 2M3S$_2$ was the highest and occurred at a comparable level to that of SM. The WSC level was higher for all intercropped conditions versus SS, but lower than that of the SM in all conditions except for 2M3S$_2$. The NDF content was the highest in SS (47.94% and 47.41% DM) and 2M2S$_2$ (47.75% DM), while there was no significant difference in the S$_1$ intercropping treatments. The 2M2S$_2$ condition produced a higher ADF content than those of the other intercropping treatments, except for 2M3S$_2$. Meanwhile, the ADF content of 2M3S$_2$ showed no significant difference compared to the rest of the intercropping treatments, and all intercropped conditions produced lower ADF compared to the SS condition.

The CP content was the highest in SS (14.60–14.79% DM), and the CP content of all intercropped treatments was higher than that of SM. The 2M2S condition showed the lowest CP content (8.10% and 7.98% DM) in intercropping treatments. The CP content of 2M4S$_2$ was higher than that of 2M3S$_2$. CP content occurred at comparable levels between 2M3S$_1$ and 2M4S$_1$. The RFV value of SS was the lowest (127.09). Among the intercropping conditions, 2M2S$_2$ exhibited the lowest RFV value, followed by 2M4S$_2$, and the highest value was observed for 2M3S$_1$ (160.20). No significant difference was found among the intercropping treatments of S$_1$.

**Table 4.** Chemical composition of maize and soybean before ensiling as affected by different planting patterns in 2018.

| Treatment | WDM % | WSC (%WDM) | CP (%WDM) | NDF (%WDM) | ADF (%WDM) | RFV |
|---|---|---|---|---|---|---|
| 2M2S$_1$ | 35.40 [c] | 15.51 [b] | 8.10 [d] | 42.78 [b] | 22.39 [c] | 155.38 [ab] |
| 2M3S$_1$ | 36.32 [b] | 13.69 [bc] | 8.69 [c] | 41.72 [b] | 21.89 [c] | 160.20 [a] |
| 2M4S$_1$ | 35.77 [bc] | 13.67 [bc] | 8.85 [c] | 42.10 [b] | 22.09 [c] | 158.41 [ab] |
| 2M2S$_2$ | 35.52 [c] | 14.56 [b] | 7.98 [d] | 47.75 [a] | 25.81 [b] | 134.02 [c] |
| 2M3S$_2$ | 37.09 [a] | 16.22 [ab] | 8.67 [c] | 41.67 [b] | 23.86 [bc] | 156.97 [ab] |
| 2M4S$_2$ | 36.44 [b] | 15.28 [b] | 9.63 [b] | 43.40 [b] | 23.13 [c] | 151.93 [b] |
| SS$_1$ | 31.67 [e] | 7.81 [d] | 14.79 [a] | 47.94 [a] | 31.39 [a] | 125.05 [d] |
| SS$_2$ | 30.34 [e] | 8.99 [c] | 14.60 [a] | 47.41 [a] | 29.67 [a] | 129.08 [cd] |
| SM | 34.38 [d] | 17.63 [a] | 6.85 [e] | 42.94 [b] | 21.03 [c] | 157.10 [ab] |
| SEM | 0.42 | 0.66 | 0.53 | 0.47 | 0.66 | 2.655 |
| V means | | | | | | |
| S$_1$ | 34.79 | 12.67 | 10.11 | 43.64 | 24.44 | 149.80 |
| S$_2$ | 34.85 | 14.26 | 10.22 | 45.06 | 25.82 | 143.02 |
| P means | | | | | | |
| 2M2S | 35.46 | 15.04 | 8.04 | 45.27 | 24.10 | 144.74 |
| 2M3S | 36.71 | 15.94 | 8.68 | 41.69 | 22.87 | 158.64 |
| 2M4S | 36.11 | 14.49 | 9.24 | 42.75 | 22.61 | 155.17 |
| SM | 34.38 | 17.63 | 6.85 | 42.94 | 21.03 | 157.13 |
| SS | 31.01 | 8.40 | 14.70 | 47.68 | 30.53 | 127.09 |
| Significance | | | | | | |
| V | <0.01 | <0.01 | 0.16 | <0.01 | 0.02 | <0.01 |
| P | <0.01 | <0.01 | <0.01 | <0.01 | <0.01 | <0.01 |
| V*P | <0.01 | <0.01 | <0.01 | <0.01 | <0.01 | <0.01 |

S$_1$, treatments containing crop soybean; S$_2$, treatments containing forage soybean; 2M2S, 2 maize rows plus 2 soybean rows strip intercropping; 2M3S, 2 maize rows plus 3 soybean rows strip intercropping; 2M4S, 2 maize rows plus 4 soybean rows strip intercropping; SS$_1$, crop soybean sole cropping; SS$_2$, forage soybean sole cropping; SM, maize sole cropping. WDM, whole crop dry matter; CP, crude protein; WSC, water-soluble carbohydrates; NDF, neutral detergent fiber; ADF, acid detergent fiber; RFV, relative feed value. V, main effect of soybean varieties; P, main effect of planting patterns; V*P, interactive effect between soybean varieties and planting patterns. SEM, standard error of means. Means are averaged over three replicates. Means that do not share the same letters in the column differ significantly ($p < 0.05$).

*3.3. Silage Quality*

As shown in Table 5, all silage chemical compositions were affected by the interaction of soybean varieties and planting patterns ($p < 0.01$), except for WSC content. The DM content was also affected by soybean varieties and the interaction of soybean varieties and planting patterns. The DM content decreased after ensiling and was higher for the intercropping treatments than the sole cropping treatments (1.23–5.28% DM), with SM exhibiting higher levels than that of SS. Moreover, with the exception of 2M2S, the DM content of S$_2$ treatments was higher than those of the S$_1$ treatments. The DM content of 2M3S was the highest (35.15% and 36.43%), but no difference was found among the S$_1$ intercropping treatments. The NDF content was highest in the 2M4S$_1$ and 2M2S$_2$ conditions (47.08% and 47.35% DM, respectively) and lowest in 2M3S (43.82% DM), whilst the NDF of SS was higher than that of SM. The ADF content showed no difference between different soybean varieties in intercropping treatments, except for the higher ADF content in 2M4S$_1$ versus that of 2M4S$_2$. The lowest ADF content was observed for 2M3S and SM, and the highest was found in the SS treatment (28.41% and 30.76% DM). The WSC content was affected by planting patterns ($p < 0.01$). It decreased after ensiling, with 2M2S exhibiting the highest level (4.70% DM) followed by 2M3S, and no difference observed among the other treatments. The 2M2S$_1$ and 2M3S$_1$ treatments had a higher CP content than that of 2M4S$_1$, and 2M2S$_2$ had lower CP content than that of 2M3S$_2$. All intercropping treatments had higher CP contents than that of SM (6.44% DM) but lower CP contents than that of SS (13.48% DM). The RFV values of 2M2S$_1$ and 2M3S$_1$ were higher than that of 2M4S$_1$ in S$_1$ intercropping treatments, while 2M3S$_2$ and 2M4S$_2$ showed higher RFV values in S$_2$

intercropping treatments. Among the different planting patterns, SM (146.45) and 2M3S (148.53) had the highest RFV values.

**Table 5.** Chemical composition of maize and soybean silage at 60 days of ensiling as affected by different planting patterns in 2018.

| Treatment | WDM % | WSC (%WDM) | CP (%WDM) | NDF (%WDM) | ADF (%WDM) | RFV |
|---|---|---|---|---|---|---|
| 2M2S$_1$ | 34.92 $^c$ | 4.98 | 7.81 $^{cd}$ | 43.79 $^d$ | 25.08 $^c$ | 147.35 $^a$ |
| 2M3S$_1$ | 35.15 $^c$ | 3.64 | 7.96 $^c$ | 43.61 $^d$ | 24.27 $^d$ | 149.30 $^a$ |
| 2M4S$_1$ | 34.38 $^{cd}$ | 2.30 | 7.46 $^e$ | 47.08 $^a$ | 25.21 $^c$ | 136.85 $^b$ |
| 2M2S$_2$ | 35.17 $^c$ | 4.41 | 7.40 $^e$ | 47.35 $^a$ | 25.82 $^c$ | 135.14 $^b$ |
| 2M3S$_2$ | 36.43 $^a$ | 3.25 | 7.73 $^{cd}$ | 44.04 $^{cd}$ | 24.35 $^d$ | 147.71 $^a$ |
| 2M4S$_2$ | 36.15 $^b$ | 2.47 | 7.58 $^{de}$ | 45.14 $^c$ | 23.64 $^e$ | 145.25 $^a$ |
| SS$_1$ | 31.27 $^e$ | 1.37 | 13.07 $^b$ | 46.47 $^b$ | 28.41 $^b$ | 133.66 $^b$ |
| SS$_2$ | 29.75 $^e$ | 1.69 | 13.90 $^a$ | 45.26 $^c$ | 30.76 $^a$ | 133.47 $^b$ |
| SM | 33.82 $^d$ | 2.98 | 6.44f | 44.28 $^c$ | 24.63 $^d$ | 146.45 $^a$ |
| SEM | 0.37 | 0.23 | 0.47 | 0.27 | 0.41 | 1.32 |
| V means | | | | | | |
| S$_1$ | 33.93 | 3.07 | 9.07 | 45.24 | 25.74 | 141.83 |
| S$_2$ | 34.38 | 2.95 | 9.15 | 45.45 | 26.14 | 140.41 |
| P means | | | | | | |
| 2M2S | 35.05 | 4.70 $^a$ | 7.60 | 45.57 | 25.45 | 141.28 |
| 2M3S | 35.79 | 3.44 $^b$ | 7.84 | 43.82 | 24.31 | 148.53 |
| 2M4S | 35.27 | 2.39 $^c$ | 7.52 | 46.11 | 24.43 | 141.07 |
| SM | 33.82 | 2.98 $^c$ | 6.44 | 44.28 | 24.63 | 146.45 |
| SS | 30.51 | 1.53 $^c$ | 13.48 | 45.87 | 29.58 | 133.61 |
| Significance | | | | | | |
| V | <0.01 | 0.60 | 0.11 | 0.53 | 0.25 | 0.28 |
| P | <0.01 | <0.01 | <0.01 | <0.01 | <0.01 | <0.01 |
| V*P | <0.01 | 0.49 | <0.01 | <0.01 | <0.01 | <0.01 |

S$_1$, treatments containing crop soybean; S$_2$, treatments containing forage soybean; 2M2S, 2 maize rows plus 2 soybean rows strip intercropping; 2M3S, 2 maize rows plus 3 soybean rows strip intercropping; 2M4S, 2 maize rows plus 4 soybean rows strip intercropping; SS$_1$, crop soybean sole cropping; SS$_2$, forage soybean sole cropping; SM, maize sole cropping. WDM, whole crop dry matter; CP, crude protein; WSC, water-soluble carbohydrates; NDF, neutral detergent fiber; ADF, acid detergent fiber; RFV, relative feed value. V, main effect of soybean varieties; P, main effect of planting patterns; V*P, interactive effect between soybean varieties and planting patterns. SEM, standard error of means. Means are averaged over three replicates. Means that do not share the same letters in the column differ significantly ($p < 0.05$).

The fermentation profiles are shown in Table 6. The results indicate that pH was affected by planting patterns ($p < 0.01$). It was highest in SS (4.37), followed by 2M4S, whilst the pH of 2M2S, 2M3S, and SM showed comparable levels. The NH$_3$-N TN$^{-1}$ was also affected by the interaction of planting patterns and soybean varieties ($p = 0.03$), with no significant differences observed between SS$_2$ and the S$_2$ intercropping treatments, but higher levels detected for SS$_1$ than for S$_1$ intercropping treatments. The LA, AA, and PA content were all affected by soybean varieties and their interaction with planting patterns ($p < 0.01$). The LA content was higher for the S$_2$ intercropping treatments than the S$_1$ intercropping treatments. The LA content of all intercropping treatments was higher than that of SS (6.32 g kg$^{-1}$) and lower than that of SM (45.77 g kg$^{-1}$). The AA content was higher for the S$_2$ intercropping treatments than for the S$_1$ intercropping treatments. The AA contents of all intercropping treatments were lower than that of SS (20.30 g kg$^{-1}$) and comparable to that of SM (5.33 g kg$^{-1}$). The PA content showed no significant difference among S$_2$ intercropping treatments or between these treatments and SM. Meanwhile, 2M3S$_1$ had the highest PA content compared to the other S$_1$ intercropping treatments. The V-score was affected by the planting patterns ($p < 0.01$), occurring at the lowest level in SS (72.51) and the highest level in 2M3S (94.10), with no significant differences found among the other treatments.

**Table 6.** Fermentation profiles of maize and soybean silage at 60 days of ensiling as affected by different planting patterns in 2018.

| Treatment | pH | $NH_3$-N $TN^{-1}$ % | LA (g $kg^{-1}$ WDM) | AA (g $kg^{-1}$ WDM) | PA (g $kg^{-1}$ WDM) | V-Score |
|---|---|---|---|---|---|---|
| 2M2S$_1$ | 3.91 | 9.54 [bc] | 20.02 [e] | 2.18 [f] | 0.94 [c] | 90.92 |
| 2M3S$_1$ | 3.81 | 7.03 [c] | 30.95 [c] | 4.19 [e] | 1.95 [b] | 95.93 |
| 2M4S$_1$ | 4.11 | 10.51 [bc] | 24.19 [de] | 5.11 [de] | 1.14 [c] | 87.95 |
| 2M2S$_2$ | 3.92 | 8.86 [bc] | 27.39 [cd] | 7.65 [c] | 2.50 [ab] | 91.68 |
| 2M3S$_2$ | 3.88 | 8.69 [c] | 36.73 [b] | 6.42 [cd] | 2.46 [ab] | 92.26 |
| 2M4S$_2$ | 3.99 | 9.15 [bc] | 36.13 [b] | 10.27 [b] | 2.77 [a] | 90.77 |
| SS$_1$ | 4.43 | 15.52 [a] | 6.41 [f] | 20.70 [a] | ND | 65.84 |
| SS$_2$ | 4.31 | 12.23 [b] | 6.23 [f] | 19.90 [a] | ND | 79.18 |
| SM | 3.90 | 9.27 [bc] | 45.77 [a] | 5.33 [de] | 2.25 [ab] | 91.45 |
| SEM | 0.39 | 0.45 | 1.22 | 1.15 | 0.19 | 1.32 |
| V means | | | | | | |
| S$_1$ | 4.07 | 10.65 | 20.39 | 8.05 | 1.00 | 85.16 |
| S$_2$ | 4.02 | 9.73 | 26.62 | 11.06 | 1.93 | 88.47 |
| P means | | | | | | |
| 2M2S | 3.85 [c] | 9.20 | 23.70 | 4.92 | 2.20 | 91.30 [b] |
| 2M3S | 3.92 [c] | 7.86 | 33.84 | 5.31 | 1.72 | 94.10 [a] |
| 2M4S | 4.05 [b] | 9.83 | 30.16 | 7.69 | 1.95 | 89.36 [b] |
| SM | 3.90 [c] | 9.27 | 45.77 | 5.33 | 2.25 | 91.45 [b] |
| SS | 4.37 [a] | 13.88 | 6.32 | 20.30 | 0.00 | 72.51 [c] |
| Significance | | | | | | |
| V | 0.33 | 0.11 | <0.01 | <0.01 | <0.01 | 0.07 |
| P | <0.01 | <0.01 | <0.01 | <0.01 | <0.01 | <0.01 |
| V*P | 0.36 | 0.03 | <0.01 | <0.01 | <0.01 | 0.23 |

S$_1$, treatments containing crop soybean; S$_2$, treatments containing forage soybean; 2M2S, 2 maize rows plus 2 soybean rows strip intercropping; 2M3S, 2 maize rows plus 3 soybean rows strip intercropping; 2M4S, 2 maize rows plus 4 soybean rows strip intercropping; SS$_1$, crop soybean sole cropping; SS$_2$, forage soybean sole cropping; SM, maize sole cropping. WDM, whole crop dry matter; LA, lactic acid; AA, acetic acid, PA, propionic acid; $NH_3$-N, ammonia nitrogen; TN, total nitrogen. V, main effect of soybean varieties; P, main effect of planting patterns; V*P, interactive effect between soybean varieties and planting patterns. SEM, standard error of means. Means are averaged over three replicates. Means that do not share the same letters in the column differ significantly ($p < 0.05$).

### 3.4. Correlation Analysis

To determine the relationship between the fermentation profiles and chemical compositions of the silages and the yield, Pearson correlation analysis was performed (Figure 3). The pH level positively correlated with the $NH_3$-N $TN^{-1}$, CP, AA, ADF ($p < 0.01$), SDM, and NDF ($p < 0.05$), and negatively correlated with the LA, PA, DM, WSC, MDM, and MSDM ($p < 0.01$). The $NH_3$-N $TN^{-1}$ positively correlated with the ADF, AA, CP ($p < 0.01$), and negatively correlated with the LA, PA, DM, WSC, MDM, and MSDM ($p < 0.01$). The LA positively correlated with the PA, DM, MDM, and MSDM ($p < 0.01$), and negatively correlated with the AA, CP, ADF, and SDM ($p < 0.01$). The AA positively correlated with the CP and ADF ($p < 0.01$), and negatively correlated with the PA, DM, WSC, MDM, and MSDM ($p < 0.01$). The PA positively correlated with the DM, MDM, MSDM ($p < 0.01$), and WSC ($p < 0.05$), and negatively correlated with the CP, ADF, and SDM ($p < 0.01$). The DM positively correlated with the WSC, MDM, and MSDM ($p < 0.01$), and negatively correlated with the CP and NDF ($p < 0.01$). The WSC positively correlated with the MDM and MSDM ($p < 0.01$), and negatively correlated with the CP and ADF ($p < 0.01$). The CP positively correlated with the ADF and SDM ($p < 0.01$), and negatively correlated with the MDM and MSDM ($p < 0.01$). The ADF negatively correlated with the MDM and MSDM ($p < 0.01$), while the positively MDM correlated with the MSDM ($p < 0.01$).

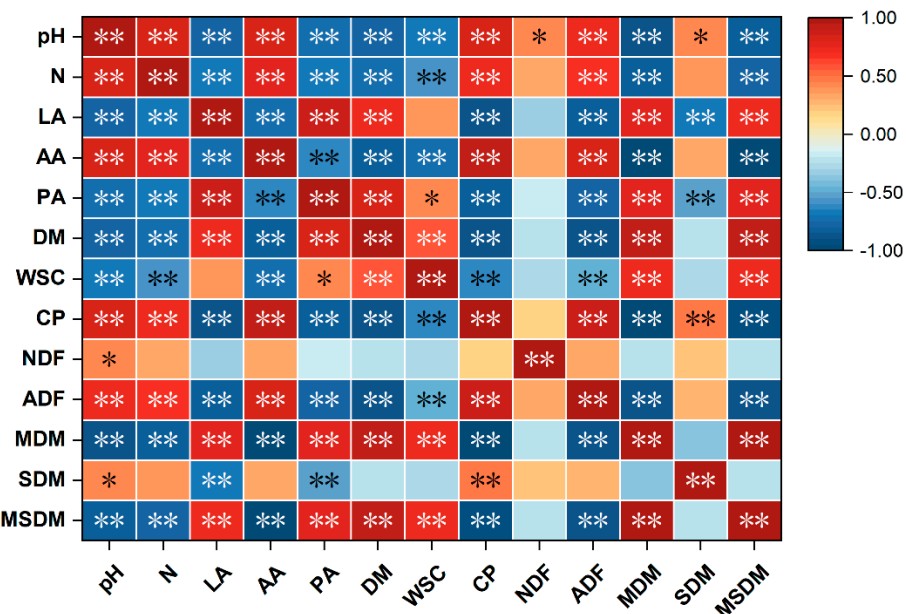

**Figure 3.** Correlation analysis between field treatments and silage quality. LA, Lactic acid content; AA, acetic acid content; PA, propionic acid content; N, $NH_3$-N/TN; DM, dry matter content; CP, crude protein; WSC, water-soluble carbohydrates; NDF, neutral detergent fiber; ADF, acid detergent fiber; MDM maize dry matter yield; SDM, soybean dry matter yield; MSDM, maize and soybean dry matter yield. The corresponding values of the heatmap are the Pearson correlation coefficient r ($-1$ to $1$), with a value below 0 representing a negative correlation (**blue**) and a value over 0 representing a positive correlation (**red**). "*" and "**" represent the relation between indices are significant at 5% and 1% level, respectively.

## 4. Discussion

### 4.1. Effect of Different Maize–Soybean Intercropping Patterns on Yield and Competition Parameters

The yield and the competition parameters were influenced by many factors [23], with the planting patterns having the greatest impact in the current study; the FM and the DM yield varied greatly with the changes in planting patterns. In both years, 2M2S had the highest MFM yield, while 2M3S had the highest MDM, MSFM, and MSDM yield, and 2M4S showed the lowest MDM and MFM yield. A similar decrease in yield corresponding to an increase in soybean row numbers was observed in a study by Raza et al. [24], who also found that a shorter distance between plants resulted in increased intraspecific competition among the maize. Moreover, the *FLERm* and *DLERm* of all SIPs in both years were around 1.00, suggesting that the SIP generally maintained maize yield at a comparable level versus sole cropping maize. Similar results regarding *LERm* were also observed in research by Liu et al. [13]. This finding can be explained as follows: maize is a high-stem crop, providing it with an edge line advantage in our wide-narrow intercropping pattern, enabling it to capture adequate solar radiation compared to sole cropping, thereby overcoming the intraspecific competition from smaller spacing among plants [25].

The highest total *FLER* and *DLER* values were found in 2M3S and 2M4S in both years, which almost doubled the land-use efficiency from the perspective of MSDM yield. A *LER* value of 1.4 in 2M3S was obtained by Raza et al. [26], which is lower than the current study, as they focused on the grain yield, in keeping with most other studies on this topic [27,28]. In this study, all total *LER* mean values among the SIPs ranged from 1.64 to 1.95, establishing that whole crop silage production utilizing this pattern offers a larger yield advantage over grain use alone.

Furthermore, the *FLERs* and the *DLERs* were always lower than one due to the lower photosynthesis of the soybean in the SIPs versus sole cropping, a result of shading by the taller maize plants [29]. Meanwhile, the corresponding *FLERm* and *DLERm* of 2M2S



and 2M3S were higher than that of 2M4S, which indicates that the field arrangements of 2M2S and 2M3S were more advantageous to maize yield. By contrast, Ren et al. [30] found no difference in *LER* values between 2M2S and 2M4S when studied under the same of planting density in intercropping and sole cropping conditions, which might be attributed to the larger densities of maize and soybean (2.4 and 1.4 times more than the current study, respectively); the edge advantage was insufficient to balance out the side effect of intraspecific competition and the growth stress caused by high density planting [31]. Likewise, *CRm* was always higher than *CRs*, indicating that maize predominated over soybean in the SIPs. The merits of maize within SIPs has also been established in a study by Ariel et al. [32], who concluded that maize possess a morphological advantage over soybean when it comes to receiving solar energy [33]. The above suggests that maize–soybean SIPs are more advantageous than sole cropping, and that 2M3S is the optimal treatment.

### 4.2. Effect of Different Maize–Soybean Intercropping Patterns on Chemical Composition of the Maize and Soybean

Different field configurations affect plant metabolism and growth due to differences in light radiation, temperature, humidity, etc. [34,35], thus influencing the chemical composition of the plants [5]. The SIP treatments had higher DM contents than those of the sole cropping treatments, with 2M3S exhibiting the highest levels, indicating that this intercropping contributed to DM accumulation. Batista et al. [14] and Erdal et al. [36] also found similar results, which might be attributed to the proper spatial arrangement of both crops within this SIP, and thus better solar radiation use efficiency [13,37]. Moreover, the SIPs generally had lower NDF and ADF contents compared to soybean sole cropping, a finding in line with Zaeem et al. [38], who found that intercropping decreased the NDF (7–9%) and the ADF (3–4%) content versus soybean sole cropping. The different WSC content between $2M3S_2$ and $2M3S_1$ might be attributable to the different soybean varieties, as they contained different WSC content in SS. The CP content increased with the increase in soybean rows, which might be due to the increased light radiation available to the soybean, a finding in line with the results of Chen [39]. All RFV values were higher than 90–115 and the standard suggested by Zaeem et al. [38]. Above all, 2M3S is more advantageous in intercropping treatments.

### 4.3. Effect of Different Maize–Soybean Intercropping Patterns on Silage Quality

Silage quality mainly depends on its fermentation and chemical composition; high DM and nutritional contents with adequate volatile acids usually indicate well-stored silage [6]. In the present study, all intercropping treatments had higher DM contents (35–36%) versus sole cropping, and the 2M3S had the highest level. Htet et al. [40] obtained a DM content of 36–41% under SIP conditions, with the higher DM content possibly being a result of their cropping patterns (1M (1, 2, 3) S); however, the fermentation was poor, as indicated by the observed pH (4.1–4.4), higher than for the present study (3.8–4.1), and it's above the indicated range for well-preserved silage, which should have a pH lower than 4.20 [41]. The $2M4S_1$ and the $2M2S_2$ treatments exhibited the highest NDF content among the SIPs and was within the standard levels (30–48%) [42]. This may have been a result of the high NDF content of $2M4S_1$ raw materials and the greatest decrease in DM content (1.92%) in this condition. Contradictory results produced by Aydemir [43] showed that the NDF and ADF content increased with the increased proportion of maize in the field, which was likely due to the higher fiber content of maize versus soybean. In contrast, the maize had lower NDF and ADF contents versus soybean in the present study. The WSC content decreased with the increase in soybean row numbers among the SIPs, of which 2M2S had the correspondingly highest WSC content. Soe et al. [44] reported that a higher percentage of soybean was associated with lower WSC content, and that increased soybean in silage also increases buffering capacity, leading to a slower pH decrease that increases the opportunity for microbes to metabolize WSC [45,46]. This finding is consistent with the present study, as the WSC content and maize yield decreased as the number of

soybean rows increased. Among the SIPs, 2M4S had the highest CP content before ensiling; however, after ensiling, the highest CP content was observed in 2M3S and $2M2S_1$. This is likely because of the relatively high AA content produced by enterobacteria degrading WSC and CP, and the high buffering capacity of soybean [47], as 2M4S had higher $NH_3$-N content and pH, indicating more proteolysis and inadequate fermentation [48]. All RFV values slightly decreased after ensiling but were still acceptable, remaining over 130 [49]; the highest RFV was found in 2M3S (around 150). Of all the SIP silages, 2M3S showed the highest DM, CP, and RFV and provides the optimal planting pattern for silage.

The fermentation quality of silage usually indicates its nutrient storage status, as better fermentation allows more nutrients to be preserved [50]. LA is the main fermentation product in silage. The $2M3S_1$ and $2M3S_2$ silages contained the highest LA concentrations and higher PA content with the same soybean varieties, which was consistent with their lowest levels of $NH_3$-N $TN^{-1}$. The $NH_3$-N $TN^{-1}$ level indicates the amount of protein degradation due to undesirable microorganisms [51,52]. LA can rapidly reduce pH value and prohibit undesirable microbes, while PA normally prohibits the growth of fungus [45,53]. The AA content increased with the increase in soybean rows, which is in accordance with the results of Parra et al. [8], who found that soybean addition favored the development of AA production by enterobacteria. The BA is usually produced by clostridia, which causes poor fermentation and an unpleasant odor [54]; no BA was detected in any of our study samples. As indicated by the V-Score, all treatments showed good fermentation quality except for soybean sole silage, and 2M3S exhibited the best level. Above all, maize–soybean mixed silage supplemented protein content without lowering the fermentation quality, which could potentially increase the quality of animal products such as milk and meat [55].

### 4.4. The Relation between Yield and Quality

The connection between yield and silage quality is indirect in the current study, and thus the correlation of the indices was analyzed. The pH, $NH_3$-N $TN^{-1}$, AA, CP, SDM, and ADF showed positive correlations with each other. Legumes usually have high buffering capacity and pH in silage [14], and a high pH usually indicates inadequate fermentation [56]. In the current study, the fermentation of soybean silage was poor; CP was degraded by undesirable microbes, and $NH_3$-N and AA were thus produced during the ensiling process [57]. In addition, higher ADF contents were found in soybean compared to that of maize. In keeping with our results, ADF has been shown to be undegradable during ensiling [58]. All the above profiles were related to soybean content, as quantified by SDM. Gao et al. [59] found similar correlations for legume silage with the exception of CP; this finding was attributed to the lower pH, which guaranteed a better fermentation, thereby preserving more CP. However, we observed contrary results in the present study, which is due to the pH lower than 4.20 in all intercropping treatments [41], meaning CP was well preserved.

The LA, PA, DM, WSC, MDM, and MSDM showed positive relations with each other. The MSDM is mainly made up of maize, and maize silage was well-fermented compared to soybean silage; therefore, more LA was produced, and more DM and WSC were perseverved. However, some propionibacteria can convert LA into PA, which might explain the positive correlation between PA and other profiles [60].

The indices of the two groups above were generally correlated negatively. Ni et al. [6] also found higher ADF, CP contents, and $NH_3$-N $TN^{-1}$ ratios with the increase in soybean addition to maize; though soybean supplies an important protein source, the quantity needs to be checked carefully to achieve the optimal balance between fermentation quality and nutritional content.

Above all, MDM and MSDM correlated with a better silage quality versus SDM, except for the CP content; the maize–soybean mixed silage is thus recommended.

## 5. Conclusions

Maize–soybean strip intercropping improved biomass yield and silage quality compared to sole cropping/silage. All SIPs had at least a 1.6 times higher biomass yield compared to sole cropping, with the 2M3S showing the highest total yield. Maize and soybean mixed silage improved nutritional value compared to maize sole silage, and fermented better than soybean sole silage, and the 2M3S provided the optimal ratio. Based on these results, maize–soybean strip intercropping is recommended to produce silage, and may offer a new way to develop a source of feedstock other than that of grain alone.

**Supplementary Materials:** The following supporting information can be downloaded at: https://www.mdpi.com/article/10.3390/fermentation8040174/s1, Table S1: Field arrangements of maize–soybean strip intercropping.

**Author Contributions:** Conceptualization, T.Z. and Y.W.; methodology, T.Z. and X.L.; software, T.Z.; validation, M.T.; formal analysis, C.C.; investigation, J.Z.; resources, X.W. and B.K.; data curation, D.J.; writing—original draft preparation, T.Z. and Y.W.; writing—review and editing, T.Z.; visualization, Z.D. and Y.X.; supervision, Y.Y.; project administration, W.Y.; funding acquisition, Y.Y. All authors have read and agreed to the published version of the manuscript.

**Funding:** This research was funded by the National Natural Science Foundation of China (32001401) and Sichuan Science and Technology Department Programs (grant numbers 2020YFN0021, 2021YFH0155, 2021YFN0059 and 2021YFQ0015).

**Institutional Review Board Statement:** Not applicable.

**Informed Consent Statement:** Not applicable.

**Data Availability Statement:** Not applicable.

**Acknowledgments:** Authors are grateful for the support of Sichuan Agricultural University.

**Conflicts of Interest:** The authors declare no conflict of interest.

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
