# Peer review of "Silage Quality and Output of Different Maize–Soybean Strip Intercropping Patterns"

_fermentation, doi:10.3390/fermentation8040174_

Round 1

Reviewer 1 Report

Research motivation is important and the issues are undoubtedly worth considering. The article is well written and the results are detailed. Chapter 2.6 needs to be improved. Basic information is missing, such as: which tests were used to check the conditions for ANOVA, what post-hoc tests were performed, etc. This information was also not included in the descriptions of figures and tables. If an interaction is found to be significant on the basis of ANOVA, what is the point of performing a post-hoc test for the effects of the main factors? For significant interactions, post-hoc testing should be performed for a combination of experimental factors and not for the main effects. The results of tests of simple main effects should be considered suggestive and not definitive. Therefore, the tables and the description of the results for variables for which the interaction is significant should be corrected.

In the description of the Figure 3 it appears that Pearson correlation coefficient r (-2 to 2). The standard correlation coefficient is in the range [-1; 1], so it is worth developing this topic in the methodology.

Editorial notes: There are two subsections with the same title 2.3.1 and 2.3.2.

Line 114: Please reword the sentence.

Reviewer 2 Report

Dear authors,

The manuscript agronomy-1637959, entitled “Silage quality and output of different maize-soybean strip intercropping patterns” presented by Tairu Zeng et al. report the results related to a two-year field experiment aimed at studying the production and silage quality of a corn-soybean intercropping grown in strips.

Considering the importance of studies on intercropping along with silage quality in order to develop new sustainable cropping and forage production systems, I believe the manuscript is of potential interest to readers of "Fermentation" and falls within its scope.

In general, the experimental activity was conducted following a rigorous scientific logic and according to widely used methods.  However, the manuscript needs some important changes to make it suitable for publication.

Below are my specific comments, which I hope will help the authors to improve the manuscript.

  1. Keywords:
  • line 28: Avoid repeating in the keywords the same words already in the title.

  1. Introduction: The introduction section needs to be implemented and enriched with more citations. It needs to deepen and present the state of the art of the research topic developed in this study and also to make it balanced with the other sections.
  • Line 37 is not well connected with the previous period. I recommend revising the sentence [6].

  1. Materials and methods: the materials and methods section is ber organised however some revisions are needed:
  • Line 65 delete double parenthesis
  • Line 67 three soybean rows (2M3S)?
  • Lines 69 - 70 in relation to sowing density I recommend that the manuscript should state the number of plants per m2 instead of the number of plants per ha.
  • Lines 73 - 76 Considering the sowing period in the two years (17 and 24 April, respectively) and the harvesting time carried out on 29 July 2018 and 10 August 2019 I request the authors to provide information on the biological cycle length (FAO Class, Group) of maize and soybean and to justify the choice.
  • Line114-117 rewrite
  • Line 119 nutritional
  • Line 122-123 rewrite
  • Line 123 according with
  • Line 123 citation style should be revised
  • Lines 136-138 were the yield data from the two experimental years analyzed separately? It is not apparent from the tables. Also, there is no reference in the text to any interaction with the year factor. Is it advisable to better explain the statistical treatment of the experimental data and to indicate which test was applied for the significance of the averages?

  1. Results: also the results section should be reviewed: in general it is suggested to reduce the size of table 1 taking into account the statistical treatment used for as requested above.
  • Line 149 among the instead of “in most of the”.
  • Line 155 this part should be moved above table 2.
  • Line 168 put the table above figure 2

  1. Discussion: the discussion of the results in principle is fine, I only suggest to modify the first part namely:
  • Line 310-316 rewrite. it is difficult to read and understand.

  1. Conclusions: in the conclusions you should not repeat the results but present your main considerations related to what you observed in the results and then discussed
  • Line 432

Round 2

Reviewer 2 Report

For Authors

Manuscript Number: Fermentation -1637959: Silage quality and output of different maize-soybean strip intercropping  patterns.

I thank the authors for making the requested changes. These have been fully addressed and have resulted in an overall improvement of the manuscript.

Regards